# NRas Nanoclusters Mediate Crosstalk Between BRAF/ERK and PI3K/AKT Signaling in Melanoma Cells

**DOI:** 10.3390/ijms262311647

**Published:** 2025-12-01

**Authors:** Oren Yakovian, Julia Sajman, Eilon Sherman

**Affiliations:** 1Racah Institute of Physics, The Hebrew University, Jerusalem 91904, Israel; 2Walder Department of Bioinformatics, Tal Campus for Women, Jerusalem College of Technology, Havaad Haleumi 21, Givat Mordechai, Jerusalem 91160, Israel

**Keywords:** melanoma, single molecule localization microscopy, super-resolution microscopy, protein clustering, Ras, PI3K, AKT

## Abstract

Melanocyte signaling through the MAPK pathway is orchestrated by NRas and relayed downstream via multiple effectors, such as RAF, Ral, and PI3K. In spite of their significance, the molecular mechanisms of signaling relay by NRas, their dynamics, and their potential as therapeutic targets remain unclear. Using multi-color single molecule localization microscopy (PALM and dSTORM), we resolved the mutual nanoscale organization of NRas, PI3K, and BRAF at the plasma membrane of fixed and live melanoma cells. Surprisingly, NRas and its oncogenic mutation Q61R colocalized with PI3K in mutual nanoclusters, where BRAF was also frequently present. In live cells, NRas and PI3K co-clustering declined, yet persisted over minutes. Clinically relevant perturbations revealed unexpected crosstalk within these nanoclusters and consequently, between the MAPK and PI3K pathways. Specifically, overexpression of the Ras binding domain (RBD) and PI3K inhibition by wortmannin disrupted NRAS-PI3K interactions, and reduced both pAKT and pERK levels and cancer cell proliferation. MEK inhibition with trametinib resulted in similar, yet more pronounced effects. Thus, our findings provide novel insights into NRAS-mediated signaling through nanoscale clusters and underscore their potential as therapeutic targets.

## 1. Introduction

Ras protein resides at the intersection between the mitogen-activated protein kinase (MAPK) and the PI3K/AKT/mTOR pathways, playing a pivotal role in regulating cell survival, growth, migration, and metabolism [1]. In stromal cells, these pathways orchestrate critical cellular processes and contribute to the intricate communication networks within the healthy tissue and the tumor microenvironment. Ras mutations can constitutively activate both pathways, leading to aberrant signaling frequently associated with tumor initiation, progression, and resistance to therapies [2,3].

Melanoma, one of the most aggressive and treatment-resistant forms of skin cancer, is characterized by its complex and dynamic molecular signaling networks. Among the key drivers of melanoma progression are mutations in the Ras family of small GTPases, particularly NRas [4], and the dysregulation of downstream signaling pathways. For instance, high levels of phosphorylated AKT (pAKT) have been observed in two-thirds of primary and metastatic melanomas [5]. Despite the importance of these pathways, our understanding of their spatial organization within cells, particularly at the nanoscale, remains limited.

Previous studies on the nanoscale organization of Ras proteins at the plasma membrane (PM) have elucidated the formation of spatially confined and dynamic nanoscale domains known as Ras nanoclusters [6]. These nanoclusters serve as signaling platforms where active Ras interacts with downstream effectors such as RAF, RalGDS, and PI3K [6,7]. The precise spatial arrangement of Ras proteins within nanoclusters regulates the strength and specificity of effector binding, thereby enhancing signaling efficiency [8]. Extensive characterization of Ras-RAF interactions within complexes (with MEK, 14-3-3, and Gal-3) and nanoclusters has been performed [9,10]. Recent advances in super-resolution microscopy and specifically in single-molecule imaging [11], have served to better characterize Ras-nucleated nanoclusters in fixed and live cells [12,13,14]. This dynamic spatial organization plays a critical role in regulating the intensity, duration, and specificity of signaling events.

PI3K activation by Ras has been established [15] and their structural interaction has been resolved [16,17,18,19]. Still, the direct role of PI3K in Ras nanoclusters remains less studied compared to other effectors, especially using advanced imaging approaches such as single molecule localization microscopy (SMLM). This gap in knowledge limits our understanding of how Ras nanoclusters contribute to PI3K/AKT pathway activation and signaling relay and fidelity in health, or aberration in cancer.

Here, we explore the nanoscale clustering of NRas and PI3K in melanoma cells and investigate how these clusters contribute to signaling fidelity and cellular behavior. Using SMLM in two and three colors, we find that NRas, PI3K, and BRAF form mutual nanoclusters at the plasma membrane, which apparently serve as signaling hubs that are essential for the activation of downstream pathways, including the PI3K/AKT and BRAF/MAPK pathways. In live cells, NRas and PI3K colocalization in the nanoclusters declined, yet persisted over minutes. We further employ clinically relevant perturbations to these pathways and reveal unexpected crosstalk within these nanoclusters and, consequently, between the MAPK and PI3K pathways. Specifically, we examine the impact of AKT inhibition on these clusters, showing that disruption of AKT activity alters the organization of NRas-PI3K nanoclusters, thereby providing insights into the regulatory mechanisms controlling melanoma cell signaling.

Taken together, our findings provide new evidence for the role of nanoscale organization in the regulation of critical signaling pathways in melanoma and suggest that targeting these nanoscale interactions should be better understood for more effective cancer therapy, for instance in the context of resistance to conventional treatments.

## 2. Results

### 2.1. Mutual Co-Clustering of NRas and PI3K at the Plasma Membrane of Melanoma Cells

PI3K is one of the main Ras effectors [20]. We employed two-color PALM in total internal reflection (TIRF) mode to study the nanoscale organization of NRas and PI3K at the PM of fixed 108T melanoma cells. Such imaging allowed us to resolve single PAGFP-NRas and PAmCherry-PI3K co-expressed in the cells with a lateral resolution ~20–30 nm (Appendix A). The two proteins were recruited to the PM (Figure 1A) with overall densities of 16.1 ± 2.15 molecules/μm^2^ and 11.1 ± 0.95 molecules/μm^2^ for NRas and PI3K, respectively (Figure 1B). Strikingly, the two proteins colocalized in pronounced nanoclusters (Figure 1A).

Self-clustering was analyzed by the pair-correlation function g(r) (Figure 1C,D) and showed × 219.3 (i.e., <g(0–20) ≥ 219.3) (NRas) and ×16.8 (i.e., <g(0–20) ≥ 16.8) (PI3K) fold extent of clustering over random Poisson distribution (i.e., g(r) = 1 for all r values). Mapping the PCF functions and densities for individual cells has been useful for elucidating changes under various conditions [13], and is thus depicted for convenience (Appendix A).

We analyzed the co-clustering of NRas and PI3K using coordinate based colocalization (CBC; see Section 4) analysis. This analysis colors each molecule of one type (here NRas, in green), based its proximity to molecules of a second type (PI3K, in red) (Figure 1E). The CBC distribution, averaged across 35 cells, showed a distinct peak at values of +1, that is also indicative of high colocalization of the NRas and PI3K [21] (Figure 1F). The smaller peak at values of +0.6 is indicative of the partial colocalization of these proteins in proximal or partially overlapping clusters [21].

PALM involves overexpression of proteins that are genetically tagged with photoactivatable fluorescent proteins, which may lead to artefacts [22]. Thus, we conducted imaging of NRas and PI3K clustering using dSTORM (in TIRF mode), which involves labeling the endogenous target proteins with fluorescently labeled antibodies and does not involve protein overexpression. As expected, NRas and PI3K resided in mutual clusters at the plasma membrane of the 108T melanoma cells (Appendix A). The extent of self-clustering and co-clustering of NRas and PI3K was similar to the PALM results (compare Appendix A with Figure 1C,D,F). Notably, the concentration of these molecules obtained through dSTORM imaging was about five times higher than in PALM (Figure 1B). A direct comparison between the measured concentrations by PALM and dSTORM is complicated due to the different cohort of measured proteins, and possible overcounting issues by dSTORM due to multiple labeling of the antibodies (carrying about three fluorophores each). Moreover, dSTORM is limited to fixed cell imaging. Thus, we focused on PALM imaging through most of the study.

### 2.2. PI3K Recruitment to NRas Clusters Declines, Yet Persists over Minutes

Next, to study the potential dynamics of the interaction between NRas and PI3K, we conducted live cell two-color PALM (TIRF) imaging of these proteins at the plasma membrane of cells, as they were dropped on coverslips under the microscope. The coverslips were precoated with PLL and EGF, to trigger cell adhesion, spreading and activation of EGF receptors. Indeed, we detected clusters of NRas and PI3K at the PM of these cells from contact detection (Figure 1A). While some dynamics of the clusters and the molecules were visible (Appendix A), these proteins showed persistent colocalization in some, but not in all the clusters (Figure 2B, warm/cold colors in jet color bar and filled/empty arrowheads indicate persistent/transient interactions in clusters, respectively). The distribution of CBC values showed a distinct peak of CBC ~1 throughout the imaging sequence.

To further quantify the dynamics of the proteins co-clustering, we consider another useful measure that we call “CBC ratio” for capturing differences in the CBC value distribution under various conditions (here, over time). This ratio is calculated as follows: we divided the distribution into four equal bins [i.e., (−1:−0.5), (−0.5:0), (0:0.5), (0.5:1)] and note that the (0.5:1) bin is most sensitive to high colocalization, while the (−1:−0.5) bean indicates high segregation [21] (insets in Figure 2C). Thus, the ratio of these beans captures differences in the colocalization vs. segregation of the proteins under study. The analysis of CBC ratio over time indicates high co-clustering of NRas and PI3K, followed by some decline yet persistent co-clustering over minutes (Figure 2D, Appendix A).

### 2.3. PI3K Co-Clusters with Oncogenic NRas Mutants

Oncogenic mutations in NRas are frequently found in melanoma cells [23]. Hence, we studied the mutual organization of wt NRas with the most frequent oncogenic mutation Q61R, found in 85% of melanoma cases incorporating NRas mutations [24]. Surprisingly, NRas-Q61R associated partially with NRas-wt, and seemed to engulf NRas-wt clusters, in contrast to the high colocalization of PAGFP-NRas-wt and PAmCherry-NRas-wt as a control (Figure 3A,F, compare zoom images on right). Still, we did not find significant differences in the densities and self-clustering of the proteins expressed in the two experiments (compare Figure 3B–D,G–I).

The differential localization of NRas-Q61R vs. NRas-wt could be further detected and quantified using CBC analyses (Figure 3E,J). The average CBC value for NRas-wt was significantly higher than the value for NRas-Q61R (*p* = 0.04), indicating again the differential localization of NRas-Q61R around NRas-wt clusters. Nevertheless, we note that the average of the complete distribution is effectively reduced by non-interacting molecules. Indeed, we find that this measure captures more sensitively the changes between the colocalization NRas-wt with itself or with NRas-Q61R (Figure 3K; *p* = 0.0267).

Our results raised the question whether NRas-Q61R recruits PI3K in a different way from NRas-wt. Thus, we imaged by two-color PALM NRas-Q61R and PI3K (Figure 3L). We found that PI3K self-clustering was significantly higher in this case, but no similar change was detected for NRas-Q61R (Figure 3M,O). Both imaging and statistical analyses (Figure 3P,Q) indicated that NRas-Q61R associated to a higher (yet not significantly different) degree with PI3K, relative to NRas-wt.

Interestingly, we have previously shown that NRas-Q61R is also effectively associated with BRAF [14]. Thus, we conclude that both wt and Q61R mutant of NRas can signal downstream through the PI3K-AKT (here) and BRAF-MAPK ([14]) pathways.

### 2.4. BRAF Recruitment to NRas Clusters Is Simultaneous with PI3K but Less Homogeneous

Signaling downstream NRas is relayed to multiple pathways, including to the MAPK pathway via BRAF, and to the AKT/mTOR pathway via PI3K. It has been proposed that the immediate NRas effectors, BRAF and PI3K, may be recruited to separate pools of NRas proteins (or clusters) [25]. Our results so far suggest, yet do not directly show, that NRas nanoclusters could serve as hubs for simultaneous recruitment of both PI3K and BRAF to the same NRas nanoclusters. Hence, we studied the mutual organization of PAGFP-BRAF and PAmCherry-PI3K using the same approach as for NRas and PI3K (i.e., as shown in Figure 1). We found that a significant fraction of BRAF and PI3K proteins closely associated in same nanoclusters (Figure 4A,E), similar to NRas and PI3K association (Figure 4F). These results further indicate the possibility of simultaneous recruitment of BRAF and PI3K to the same NRas nanoclusters.

To further verify this joint recruitment, we turned to three-color SMLM imaging of these proteins. Such imaging required the combination of two-color PALM (of BRAF-PAGFP and PI3K-PAmCherry) and dSTORM imaging of the third color (NRas labelled with Alexa647). Strikingly, we could find nanoclusters with all three proteins present (Figure 4F). The distribution of BRAF and PI3K molecules within the NRas nanoclusters seemed homogeneous, lacking internal organization (zoom images). We next employed the DBSCAN algorithm to study the mutual recruitment of these proteins in each cluster. We employed the algorithm to cluster all molecules (regardless of their type) using two threshold distances of 80 nm and of 40 nm. The 80 nm threshold preserved the observed clusters (Figure 4H and Appendix A, as in the zoom images of Figure 4G), while the 40 nm threshold broke them into smaller clusters (Appendix A). Next, we counted the number of molecules for each protein type within the mutual clusters (Figure 4G). NRas correlated very highly with PI3K using both thresholds (R_80nm_^2^ = 0.9629, R_40nm_^2^ = 0.8748), indicating homogeneous and efficient recruitment of PI3K to NRas clusters (Appendix A top panels). BRAF correlated highly with both NRas and PI3K using the 80 nm threshold (R_80nm_^2^ = 0.73) (Appendix A, middle and bottom panels), yet more moderately using the 40 nm threshold (R_40nm_^2^ = 0.3335 and 0.4132; Appendix A, middle and bottom panels). These scatter plots of protein abundance and lower correlations show that BRAF is recruit to some NRas (and PI3K) clusters more abundantly than to others (as also observed in Figure 4F). Pearson correlations between every pair of proteins under study are specified in Appendix A and support these conclusions.

We conclude that PI3K is recruited to NRas clusters more homogeneously than BRAF; i.e., effectively all NRas clusters contain PI3K, yet only a part of these clusters contains BRAF.

### 2.5. RBD Mediates Effective Recruitment of Effectors to NRas Nanoclusters

The interaction between Ras and PI3K is mediated by the Ras binding domain (RBD) in the catalytic subunit of PI3K. GTP-bound Ras interacts with p110α, potentially inducing conformational changes that stimulate PI3K catalytic activity [19,20]. BRAF also binds NRas via a slightly different RBD, which has been often used to study the Ras-RAF interactions (e.g., [26]). Thus, we next studied whether the RBD also mediates the recruitment of PI3K and BRAF to NRas nanoclusters. For that, we expressed PAGFP-RBD (of BRAF) and PAmCherry-NRas in 108T melanoma cells, imaged them, and analyzed their mutual organization using the same approach. RBD was recruited to the PM (Figure 5A,B), significantly clustered there (Figure 5A,C and Appendix A), and efficiently associated with NRas nanoclusters (Figure 5A,D,E). CBC analysis showed high colocalization of RBD and NRas (Figure 5E). We conclude that RBD mediates the recruitment of the effectors PI3K and BRAF to NRas nanoclusters.

### 2.6. RBD Overexpression Competes with PI3K Recruitment to NRas Nanoclusters

These results raised the possibility that RBD overexpression could effectively compete with the recruitment of PI3K (and of BRAF) to NRas, and thus inhibit their effector functions for potential therapeutic purposes. To study this possibility, we next expressed PAGFP-NRas and PAmCherry-PI3K in 108T melanoma cells, with and without the overexpression of RBD (Figure 5F–K). We found that RBD overexpression reduced PI3K recruitment to the PM by a factor of ~2 (Figure 5G, as compared to Figure 1B, *p* = 0.0036; see also Appendix A). Using CBC analysis (Figure 5J) and its comparison to the analysis for NRas and PI3K (Figure 5K), we found that the distributions of NRas interactions with PI3K were now significantly reduced by RBD overexpression (Figure 5K; *p* = 0.0267).

### 2.7. PI3K and NRas Self-Clustering and Co-Clustering Are Reduced by Inhibiting PI3K Activity Using Wortmannin

So far, we have shown that PI3K is effectively recruited to NRas nanoclusters. Previously, we found that NRas nanoclusters and their recruitment of downstream effectors could be disrupted by small molecule drugs [14]. Thus, we wanted to test whether PI3K recruitment to NRas clusters would be affected by inhibition of PI3K activity. For that, we imaged PAGFP-NRas and PAmCherry-PI3K in cells upon treatment with wortmannin, a potent and irreversible inhibitor of PI3K [27]. We observed reduced self-clustering of both NRas and PI3K under this condition (Figure 5N,O). While PI3K was still efficiently recruited to the NRas nanoclusters (Figure 5L), the CBC analyses showed a significant reduction in mutual association of PI3K and NRas in the nanoclusters in the wortmannin-treated cells (see Figure 5Q, which compares results in Figure 5P and Figure 1F; *p* = 0.01).

We conclude that both competition for PI3K recruitment to NRas (through RBD overexpression) and functional disruption (via wortmannin treatment) reduce its localization to NRas nanoclusters.

### 2.8. pERK Is Reduced by Cells Treatment with PI3K Inhibitor (Wortmannin) and RBD Overexpression

Taken together, our results so far raise the possibility that NRas nanoclusters could mediate significant crosstalk between the BRAF-mediated MAPK pathway and the PI3K-mediated AKT pathway, with important implications in health and disease. To study such crosstalk in signaling between the MAPK and AKT pathways, we examined whether the treatment of melanoma cells with small molecule inhibitors of one pathway could interfere with the other (Figure 6A,B and Appendix A, respectively for the two pathways). Specifically, we measured pERK in the 108T cells under the same treatment conditions (Figure 6C,D), namely: (i) wortmannin (PI3K inhibitor), (ii) trametinib (MEK inhibitor), and (iii) RBD overexpression. Controls included untreated cells (UT) and mock transfections (PEI). We found that pERK levels were significantly diminished by RBD overexpression, and by treatment with either wortmannin or with trametinib. Since 108T cells express a wt NRas and an NF1 mutation that leads to loss of regulation of NRas, these results could not be simply explained by the genetic background of these cells.

### 2.9. pAKT Is Reduced by Cells Treatment with MEK Inhibitor (Trametinib) and RBD Overexpression

Next, we measured pAKT levels in the 108T cells under the same conditions as above. We observed qualitatively that RBD overexpression diminished pAKT levels in the cells (Appendix A). Strikingly, trametinib inhibited pAKT levels with comparable inhibition efficiency to that of wortmannin (Appendix A). These results indicate a crosstalk between the MAPK and AKT pathways. They suggest that signaling within NRas nanoclusters may be synergistic, as intact PI3K signaling appears to depend on functional MAPK signaling.

### 2.10. RBD Overexpression and MEK Inhibition Significantly Decrease Melanoma Cell Growth and Induce Apoptosis

To test the functional outcome of the specified perturbations on melanoma cell growth, we treated the cells with either (1) RBD or control vector overexpression, (2) treatment with trametinib or (3) wortmannin, or (4) both drugs together. Following cell seeding, plates containing multiple replicates of each condition were treated with MTT and scanned using a plate reader on days 1, 2, 3, 6, and 8. RBD overexpression reduced cell viability as compared to overexpression of control vector in a transient fashion (Figure 6C, most significantly on day 3, *p* = 0.0136). Treatment with trametinib reduced cell viability more dramatically, while reduction due to wortmannin treatment was more modest, with little additive effect (Figure 6D). We conclude that treatment conditions that abrogated pERK and pAKT also resulted in reduction in melanoma cell viability, yet with varying efficacy and dynamics. This can be explained by the transient nature of protein overexpression in the cells, different drug efficacies, and the different roles of MAPK and PI3K on cell viability, as MAPK promotes cell growth while PI3K promotes cell survival.

### 2.11. MEK or PI3K Inhibition Induces Melanoma Cell Apoptosis

We hypothesized that the reduced growth of the melanoma cells upon treatment with wortmannin or trametinib was due to induced cell apoptosis. To test this hypothesis, we seeded 108T cells prior to the experiment in imaging chambers (NBT #81816) (25,000 cells/well). The next day, cells were treated with inhibitors and stained with Incucyte^®^ Caspase-3/7 Green Dye (Sartorius, 4440)—a marker for their caspase activity. We employed large-scale (×10) imaging of the cells and analyzed the fraction of apoptotic cells on consecutive days following the drug treatment (Figure 6E,F). A significant fraction of apoptotic cells appeared already on the first day, after 5 h (Figure 6F), and increased over time, in accordance with the results of the proliferation assay (Figure 6D).

## 3. Discussion

In this study, we applied two-color PALM and a combination with dSTORM (all in TIRF mode) to resolve the mutual nanoscale organization of NRas, PI3K and BRAF at the PM of 108T melanoma cells. NRAS and its oncogenic mutation Q61R colocalized with PI3K in mutual nanoclusters, where BRAF was also frequently present. Interestingly, we have previously shown that NRas-Q61R is also effectively associated with BRAF [14]. Thus, we conclude that both wt and Q61R mutant of NRas can serve as hubs for downstream signaling through both the PI3K-AKT (here) and BRAF-MAPK ([14]) pathways.

The interaction between NRas and PI3K declined over minutes yet persisted despite the dynamics of the molecules and clusters (Appendix A). This stands in contrast to the interaction of NRas and BRAF, whose recruitment to the nanoclusters seemed less persistent: the recruitment of BRAF proteins to the mutual clusters could either precede or supersede NRas [14]. Also, no apparent inner structure was identified in these clusters, as found for clusters of some other signaling proteins [28,29].

We employed RBD overexpression to demonstrate its incorporation into NRas nanoclusters, and then to perturb the incorporation of PI3K and of BRAF into such nanoclusters. This perturbation further resulted in reduced pAKT and pERK levels and inhibited cell proliferation. Cells treatment with clinically relevant small molecule drugs revealed unexpected crosstalk between the MAPK and PI3K pathways, induced apoptosis, and inhibition of cell proliferation, which could stem from association of BRAF and PI3K in the same NRas nanoclusters. Indeed, we previously showed that cell treatment with trametinib diminished the density of NRas and BRAF at the PM, increased their extent of self-clustering (observed also in [30]), and diminished the overlap of the self-clusters of NRas and BRAF [13]. Here, we further show that wortmannin treatment reduces NRas and PI3K self-clustering and co-clustering.

Multiple cross-talk mechanisms between MAPK and PI3K/AKT signaling have been suggested [31], and potentially include cross-inhibition, cross-activation, negative feedback loops, and pathway convergence by signaling proteins acting on the same complex or effector proteins. Still, we propose here another type of crosstalk in which the pathways signals stem from the same protein complexes (Figure 7A). Notably, such a mechanism has been shown in other signaling pathways (e.g., in complexes formed by the adapter protein LAT in T cells [32]), but not for Ras-nucleated complexes. Thus, our findings suggest that nanoclusters are a conserved mechanism for organizing signaling molecules to ensure precise and efficient signal relay. The ability of nanoclusters to coordinate multiple effectors simultaneously underscores their relevance in diverse cellular functions and contexts. Specifically, our suggested mechanism is highly relevant to the other Ras isoforms, including HRas and KRas, since they also signal through the MAPK and PI3K/AKT pathways and reside in nanoclusters [33].

The implications of our results may be considered to both monotherapy and dual therapy of melanoma (and possibly, additional cancers). First, signaling by the Q61R oncogenic NRas mutation could lead to overactivation of both Ras-ERK and PI3K-mTOR signaling, and could theoretically facilitate the development of resistance to therapeutics targeting only one pathway. The crosstalk inhibitory effect (Figure 7B), especially of trametinib on AKT signaling, may provide a delaying mechanism of such resistance development. Second, recent PI3K inhibitors (e.g., alpelisib or buparlisib; with better pharmacokinetics and reduced toxicity than wortmannin) are often investigated in melanoma in combination with BRAF inhibitors (e.g., vemurafenib) or MEK inhibitors (e.g., trametinib) to overcome resistance and improve treatment efficacy [34]. Our proliferation assay results show a minor additive effect of using both wortmannin and trametinib, since trametinib seems very potent in reducing both pERK and pAKT levels, as well as the reduction of melanoma cell proliferation.

To conclude, our findings provide novel insights into (N)Ras-mediated signaling through nanoscale clusters. We further propose that these nanoclusters could serve as novel targets for cancer therapy.

## 4. Materials and Methods

**Cell lines.** Our study was conducted on patient-derived 108T (NF1-H1366Q) and 12T melanoma cell-lines, which were received from the Samuels lab, and available from previous studies [13,14]. These cell lines were originally derived from a panel of pathology-confirmed metastatic melanoma tumor resections. Pathology-confirmed melanoma cell lines were derived from mechanically or enzymatically dispersed tumor cells, which were then cultured in RPMI1640 + 10% FBS at 37 °C in 5% CO_2_ for 5–15 passages. All cell lines have been tested negative for *Mycoplasma*.

**Plasmids.** Gateway™ pcDNA™-DEST53 Vector, 12288015, (Thermo Fisher Scientific, Waltham, MA, USA) was modified by to express PAmCherry or PAGFP instead GFP tag. The following constructs were constructed using Gateway cloning system and The RAS Clone Collection was a gift from Dominic Esposito (Addgene kit #1000000070 and kit #1000000089, Addgene, Watertown, MA, USA) (see Appendix A):

PAmCherry—NRas-WT

PAmCherry—NRas-Q61R

PAmCherry—PI3K R5 subunit

PAGFP—NRas

PAGFP—BRAF

Flag—BRAF Ras Binding Domain (RBD) [35]

PAGFP—RBD

Control vector expressing YFP

The cloning of the fluorescent tags was performed at the N-terminus of the protein to prevent localization disruption.

**Cell transfection.** The 108T melanoma cells were transfected with either two or three plasmids (depending on the experiment) using Lipofectamin 3000 (L3000008, Invitrogen, Carlsbad, CA, USA) and analyzed after 48 h. PEI transfection was also used in some experiments. Briefly, 1.5 µL of PEI (408727 Merck, Rahway, NJ, USA) per 1 µg of DNA was mixed by vortexing, incubated 10 min at RT, and then added dropwise to the cells. The next day, the medium was replaced. Cells were analyzed after 24–48 h.

Cell seeding (30,000 cells/well) and imaging was conducted on glass coverslips (#1.5 glass chambers, Cat.No:80826, ibidi GmbH, Gräfelfing, Germany), and fixed with 2.4% paraformaldehyde for 30 min at 37 °C.

**Cell staining for STORM.** For direct STORM (dSTORM) imaging, cells were permeabilized by adding 0.4 mL 0.1% Triton X-100 in PBS per well and incubated for 3–4 min. The cells were blocked by 2% normal goat serum in PFN buffer (phosphate-buffered saline, 10% fetal calf serum, and 0.02% NaN3) for 30 min. For 0.5 million cells, we added 0.5 mg mouse anti-human NRas Santa Cruz Biotechnology Cat# sc-31, RRID:AB628041, Dallas, TX, USA) and rabbit anti-human PI3K (Abcam, AB191606, Cambridge, UK) as primary antibodies diluted in 2% normal goat serum in PFN, incubated for 60 min at RT and washed three times with PFN. Secondary Alexa647 anti-mouse (Invitrogen, A21240) and Atto 488 (anti-mouse, Sigma, 62197, Merck, Rahway, NJ, USA) was added (1/3000) in 2% normal goat serum in PFN, incubated for 45 min at RT and washed three times with PFN.

**Imaging**. We performed two-color photoactivated localization microscopy (PALM) imaging following protocols established in previous studies (e.g., in [13]). Imaging was conducted using a Nikon total internal reflection fluorescence (TIRF) microscope (Nikon Instruments, Amsterdam, The Netherlands), equipped with a CFI Apo TIRF 100× oil immersion objective (numerical aperture 1.49, working distance 0.12 mm). Photoactivation of PAGFP and PAmCherry was achieved using a 405 nm laser, with intensity gradually increased from 0.5% to 10% of its 30 mW maximum output over the course of imaging. Excitation was alternated between a 488 nm laser (100% of 90 mW maximum) for PAGFP and a 561 nm laser (100% of 90 mW maximum) for PAmCherry [36]. Direct STORM imaging was conducted using 488 nm or 647 nm laser excitations (90% and 5% of the specified lasers, respectively). The three-color SMLM (PALM and dSTORM) imaging was conducted sequentially, first imaging PAGFP, next PAmCherry, and last Alexa647.

Live cell imaging was conducted on coverslips precoated with Poly-L-Lysine (PLL) and Epidermal Growth Factor (EGF; 100 ng/mL for 1 h at 37 °C). The cells were dropped on these coverslips under the microscope and imaging started (at 40 ms/frame) as soon as the cells showed contact under TIRF illumination.

All laser illuminations covered a circular area with an 80 µm diameter at the sample plane. For imaging, we acquired movies consisting of 2000 frames at 13.1 frames per second using an an Andor (Belfast, UK) iXon+ EMCCD camera. The microscope’s Perfect Focus system was employed to maintain focus throughout the imaging sessions. 

**Analyses.** We used the ThunderSTORM software (version 1.3; RRID:SCR_016897) [37] to analyze and generate PALM images. Briefly, this software served to identify individual peaks and to assign them to individual molecules for rendering of the PALM images. The localization uncertainties of the different fluorophores used in our study peaked at approximately 20–30 nm (Appendix A). A distance threshold and a temporal gap were employed for peak grouping to account for possible molecular blinking [11]. Temporal gaps were tested for each fluorophore separately to minimize possible over-counting of molecules as disc. The threshold values of ‘maximum grouping distance’ and the ‘maximum off frames’ were defined as shown in our previous study [13].

We note that SMLM (PALM and dSTORM) imaging may carry multiple complications. Most importantly, in PALM, the proteins baring fluorescent tags were over-expressed in the cells. Also, PALM and dSTORM imaging may restrict the ability to provide absolute counts of protein copy numbers. Thus, our results should be best regarded as rough estimates of protein levels, or as relative between the different imaging conditions.

**Clustering analyses.** To classify points into clusters, we used the published DBSCAN algorithm using the Matlab function “range search”(R2017b, MathWorks) [38], with a distance threshold of <80 nm or <40 nm between molecules in same cluster. This algorithm served to generate the plots that describe the molecular content of the clusters (in copy numbers for each species) [39]. See further details in Appendix A.

**Second order statistics.** A detailed description of the second-order statistics, namely the univariate pair correlation function (PCF) used in this study, has been described previously [40]. Briefly, PCF [denoted here also as g(r)] describes and quantifies in a point pattern how density varies as a function of distance from a reference particle/point. Usually, PCF is normalized by the density of the sample and was used in this normalized form throughout the study. The univariate PCF is used to explore a point pattern of a single species. It is further useful to compare the PCF results with a Poisson model that describes random placement of points across the field. This model results in a flat PCF, for which g(r) = 1. Thus, higher values of the PCF [i.e., g(r)] indicate self-clustering.

**Coordinate-based colocalization (CBC) analysis.** This analysis quantifies the spatial association between two biomolecular species by evaluating the proximity of their individual coordinates obtained from super-resolution microscopy. This method involves calculating pairwise distances between localizations of different species and determining the extent of colocalization based on a predefined distance threshold. We used an own code in Matlab (RRID:SCR_001622) (following [21]). We summarized the CBC analyses in several ways. (i) localization maps of proteins (especially of NRas) with coloring based on their individual CBC values. (ii) Distributions of CBC values. These distributions were obtained for each cell and then averaged across multiple cells. (iii) We divided the CBC distributions into coarse bins [(−1:−0.5); (−0.5–0); (0–0.5); (0.5–1)]. We then calculated a measure of ’CBC ratio’ by dividing the height of the 4th bin (colocalized molecules relative to a Poisson process) with the 1st bin for each cell (molecules that are segregated, or anti-correlated relative to a Poisson process). i.e., CBC Ratio = (# of CBC values in (0.5, 1))/(# of CBC values in (−1:−0.5)).

**Inhibitors treatment:** Cells were treated with 10 nM trametinib (S2673, Selleck Chemicals, Houston, TX, USA) or with 3 nM wortmannin (HY-10197, MedChemExpress, Monmouth Junction, NJ, USA).

Western Blot: Cells (300,000 cells/well) were treated as described in the experimental protocol. Proteins were extracted using RIPA buffer (Merck, R0278), and their concentrations were determined with the Bradford reagent (Merck, B6916). Protein extracts were separated on Novex WedgeWell 4–12% Tris-Glycine Gels (Invitrogen, XP04120BOX) and transferred onto iBlot2 NC Regular Stacks (Invitrogen, IB23001).

Antibodies used in Western blot:

Rabbit monoclonal anti-Human PhosphoERK1 (T202) + ERK2 (T185), Abcam, AB-ab201015, RRID: AB_2934088

Rabbit monoclonal anti-Human pAKT (S473), Cell Signaling Technology, Danvers, MA, USA, CST-4060S, RRID:AB_2315049

Mouse monoclonal anti-Human Tubulin (B512), Thermo Fisher Scientific, Waltham, MA, USA, 32-2500, RRID:AB_2533071

**Proliferation assay.** Proliferation assay was performed using the MTT assay of Abcam (AB-ab211091) according to the manufacturer’s protocol.

**Apoptosis assay.** To assess the apoptosis of melanoma cells, 108 T cells were seeded prior experiment in 18 well ibidi chambers (NBT #81816) (25,000 cells/well). The next day, cells were treated with inhibitors and stained with Incucyte^®^ Caspase-3/7 Green Dye (4440, Sartorius, Göttingen, Germany). Imaging was performed after 5 h, 1 day, 2 days, and 5 days. The cells were imaged using brightfield and fluorescence microscopy with X10 (air) magnification. Cell images were analyzed using Trainable Weka Segmentation (v3.3.1) in ImageJ 1.54p (NIH) to define the fraction of apoptotic cells in each field.

**Statistical significance.** The number of collected samples was selected to achieve statistical significance in the different experiments.

**Data availability.** Data supporting the findings of this study are available from the corresponding author upon reasonable request. No special code was developed for this study.

## Figures and Tables

**Figure 1 ijms-26-11647-f001:**
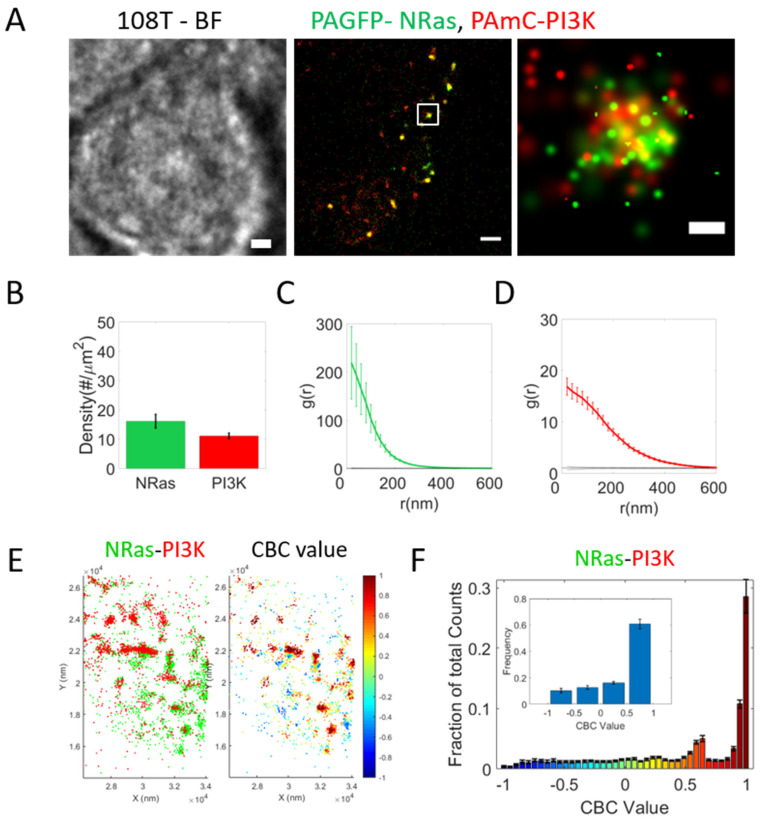
NRas and PI3K co-cluster at plasma-membrane of melanoma cells. (**A**) Brightfield and two-color PALM (TIRF) imaging of resting (fixed) 108T melanoma cells expressing PAGFP-NRas and PAmCherry-PI3K. Shown is a representative cell (*n* = 35). Zoom of middle image is shown on right. Bars, 2 µm (**left**, **middle**) and 200 nm (**right**). (**B**) Density (molecular count per area) of BRAF and NRas at the PM. (**C**,**D**) PCF of PAGFP-NRas (**C**) and of PAmCherry-PI3K (**D**). (**E**) Coordinate-based colocalization (CBC) analysis of a representative cell. Single NRas and PI3K proteins obtained by two-color PALM imaging are shown (on **left**), and NRas proteins colored by their CBC value (on **right**). (**F**) Distribution of CBC values of NRas, as analyzed in (**E**), averaged across multiple cells (*n* = 35). Inset shows distribution with coarse binning.

**Figure 2 ijms-26-11647-f002:**
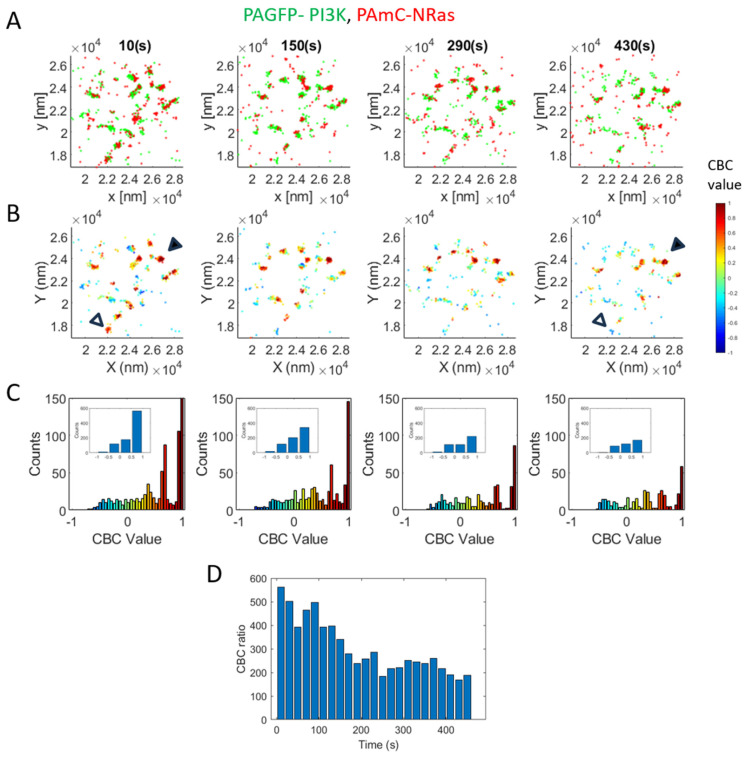
Declining yet persistent NRas and PI3K co-clustering over time. (**A**) Two-color PALM (TIRF) imaging of live 108T melanoma cells expressing PAGFP-PI3K (green) and PAmCherry-NRas (red). Cells were dropped on coated with PLL and EGF under the microscope and imaged from contact identification. A representative cell (*n* = 9) is shown at different timepoints along imaging. (**B**) CBC analysis of effective frames of imaged cell in panel (**A**). PI3K proteins are colored by their CBC value (i.e. their proximity to NRas). In left and right panels, filled and empty arrowheads exemplify clusters with persistent or transient NRas-PI3K interactions, respectively. (**C**) Distribution of CBC values of PI3K, as analyzed in (**B**). Inset shows distribution with coarse binning. (**D**) CBC ratio over time.

**Figure 3 ijms-26-11647-f003:**
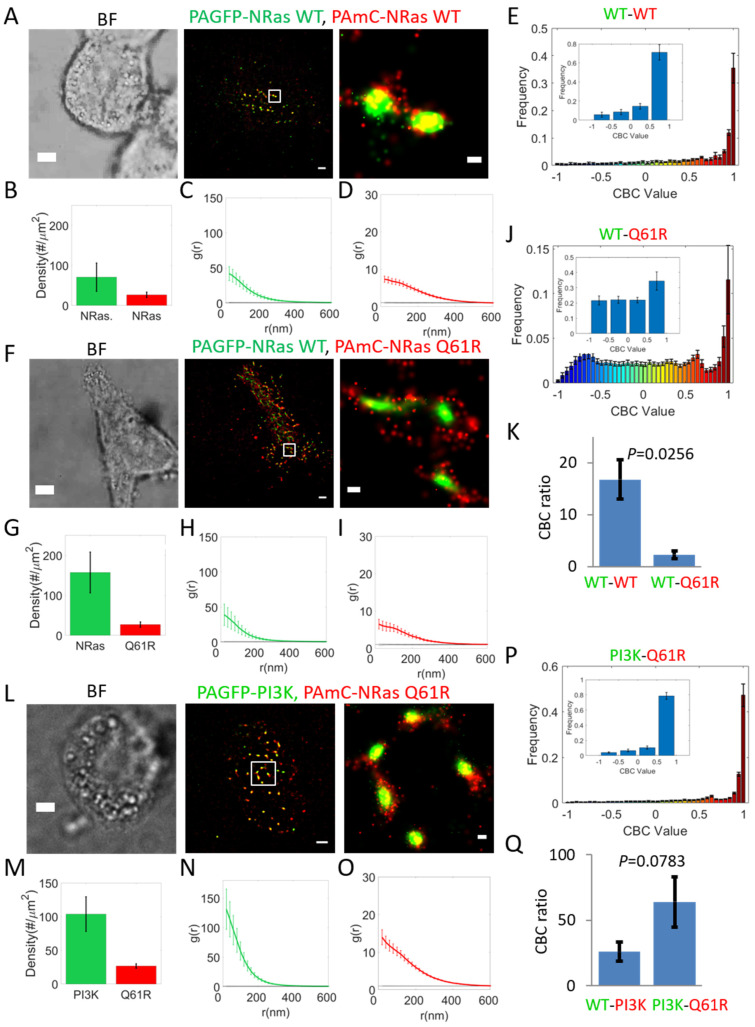
Oncogenic Q61R NRas segregates from wt NRas yet still recruits PI3K. (**A**–**E**) Relative organization of PAGFP-NRas and PAmCherry-NRas as a control. (**A**) Brightfield and two-color PALM (TIRF) imaging of resting 108T melanoma cells expressing PAGFP-NRas and PAmCherry-NRas. Shown is a representative cell (*n* = 8). Zoom of middle image is shown on right. Bars, 2 µm (**left**, **middle**) and 200 nm (**right**). (**B**) Density (molecular count per area) of PAGFP-NRas and PAmCherry-NRas at PM. (**C**,**D**) PCF of PAGFP-NRas (**C**) and of PAmCherry-NRas (**D**). (**E**) Distribution of CBC values of NRas, averaged across multiple cells (*n* = 6). Inset shows distribution with coarse binning. (**F**–**J**) Relative organization of NRas-wt and NRas Q61R (*n* = 21 cells). Imaging, analyses and scale bars as in panels (**A**–**E**). (**K**) Comparison between interaction of PAGFP-NRas with PAmCherry-NRas and with PAmCherry-NRas-Q61R. (**L**–**P**) Relative organization of PI3K and NRas Q61R (*n* = 15 cells). Imaging, analyses and scale bars as in panels (**A**–**E**). (**Q**) Comparison between interaction of PAGFP-PI3K with PAmCherry-NRas-wt (from Figure 1F) and with PAmCherry-Q61R.

**Figure 4 ijms-26-11647-f004:**
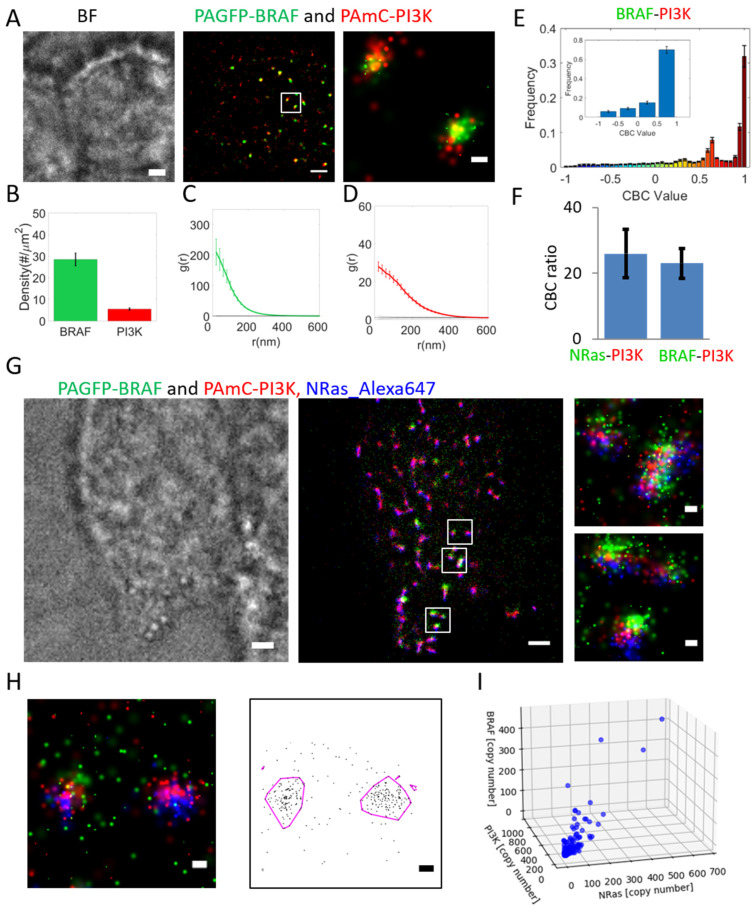
BRAF co-clustering with PI3K and with NRas. (**A**–**F**) Relative organization of PAGFP-BRAF and PAmCherry-PI3K. (**A**) Brightfield and two-color PALM (TIRF) imaging of resting 108T melanoma cells expressing PAGFP-BRAF and PAmCherry-PI3K. Shown is a representative cell (N = 39). Zoom of middle image is shown on right. Bars, 2 µm (**left**, **middle**) and 200 nm (**right**). (**B**) Density (molecular count per area) of BRAF and NRas at the PM. (**C**,**D**) PCF of PAGFP-BRAF (**C**) and of PAmCherry-PI3K (**D**). (**E**) Distribution of CBC values of BRAF, averaged across multiple cells (N = 39). Inset shows distribution with coarse binning. (**F**) Comparison between interaction of PAmCherry-PI3K with PAGFP-NRas (from Figure 1F) or with PAGFP-BRAF. (**G**) Brightfield and three-color single molecule localization microscopy (PALM and dSTORM, in TIRF) imaging of resting 108T melanoma cells expressing PAGFP-BRAF and PAmCherry-PI3K and stained with anti-NRas antibody (see Section 4). Bars, 2 µm (**left**, **middle**) and 200 nm (**right**). (**H**) Analysis of content of mutual clusters of NRas, BRAF, and PI3K (left image is a zoom of the upper white square in (**G**)), as detected by DBSCAN (**right** image; clusters, as engulfed by magenta lines, were determined using a threshold distance of 80 nm; see Section 4 and Appendix A for further details and analyses). Bars, 200 nm (**left**, **right**). (**I**) Analysis of content of mutual clusters, as imaged in (**G**) and detected in (**H**).

**Figure 5 ijms-26-11647-f005:**
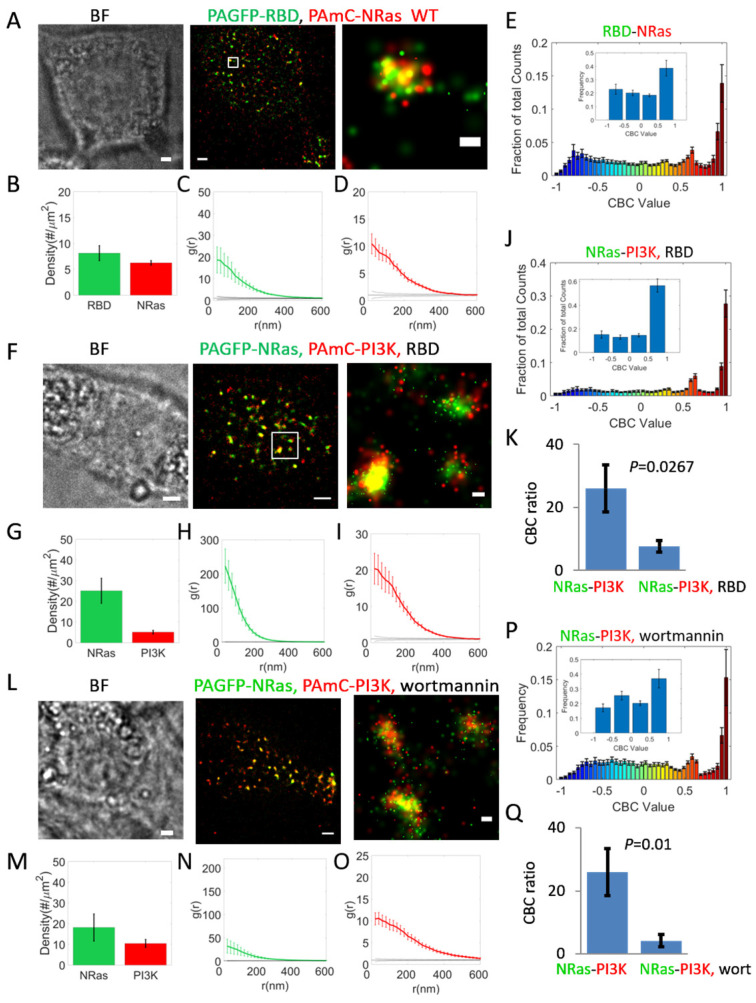
Mechanism of PI3K recruitment to NRas nanoclusters. (**A**–**E**) Relative organization of PAGFP-RBD and PAmCherry-NRas. (**A**). Brightfield and two-color PALM (TIRF) imaging of resting 108T melanoma cells expressing PAGFP-RBD and PAmCherry-NRas. Shown is a representative cell (*n* = 22). Zoom of middle image is shown on right. Bars, 2 µm (**left**, **middle**) and 200 nm (**right**). (**B**) Density (molecular count per area) of RBD and NRas at the PM. (**C**,**D**). PCF of PAGFP-RBD (**C**) and of PAmCherry-NRas (**D**). (**E**) Distribution of CBC values of NRas, as analyzed in (**E**), averaged across multiple cells (*n* = 22). Inset shows distribution with coarse binning. (**F**–**J**) Relative organization of NRas and PI3K in presence of unlabeled RBD (*n* = 21 cells). Imaging, analyses and scale bars as in panels (**A**–**E**). (**K**) Comparison between interaction of PAGFP-NRas with PAmCherry-PI3K in cells that either overexpress RBD (panel (**J**)) or not (Figure 1F). (**L**–**P**) Relative organization of NRas and PI3K in cells that were treated with 3 nM wortmannin for 24 h (N = 14 cells). Imaging, analyses and scale bars as in panels (**A**–**E**). (**Q**) Comparison between interaction of PAGFP-NRas with PAmCherry-PI3K in cells that were either treated with wortmannin (panel (**P**)) or not (Figure 1F).

**Figure 6 ijms-26-11647-f006:**
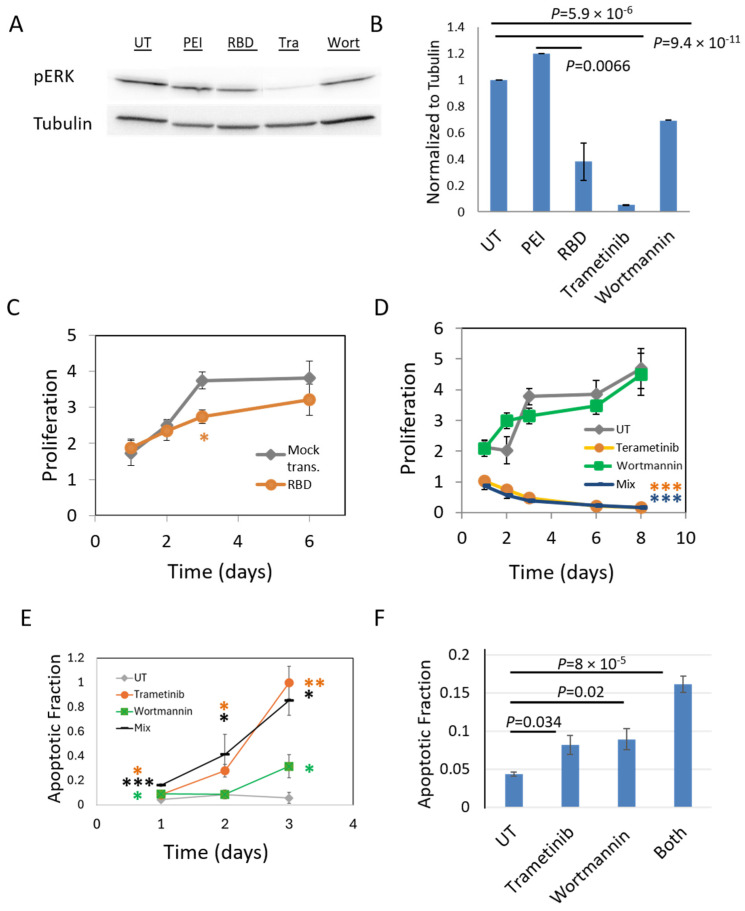
Inhibition of cell signaling and proliferation by RBD overexpression and small molecule drugs. (**A**,**B**). Western blots of resting 108T melanoma cells. Wells are shown as follows: ‘UT’—untransfected cells, ‘PEI’—mocked transfected cells, ‘RBD’—cell transfected with Flag-RBD vector, ‘Tra’—cell treated with 10 nM trametinib, ‘Wort’—cell treated with 3 nM wortmannin. Blots were stained pERK. Shown are sections of a representative blot (**A**) and quantification after calibration with tubulin for triplicate blots (**B**). (**C**,**D**) Proliferation assay of cells overexpressing either RBD or a control vector (**C**), or after treatment with either trametinib or wortmannin (same concentrations as in panels (**A**,**B**)), or with both. After seeding cells, plates with multiple repeats of each condition were treated with MTT and scanned with a plate reader on specified consecutive days (seeding means ‘day 0’). (**E**,**F**) Apoptosis assay of cells after treatment with either trametinib or wortmannin (same concentrations as in panels (**A**,**B**)), or with both. After seeding the cells, they were treated with inhibitors and stained with Incucyte^®^ Caspase-3/7 Green Dye (Sartorius, 4440). Imaging was performed after 5 h, 1 day, 2 days, and 5 days (seeding means ‘day 0’). Cells were imaged using brightfield and fluorescence microscopy with X10 (air) magnification. *, **, *** indicate *p* < 0.05, *p* < 0.01, and *p* < 0.005, respectively, of data compared to controls (i.e., mock transfection or UT cells).

**Figure 7 ijms-26-11647-f007:**
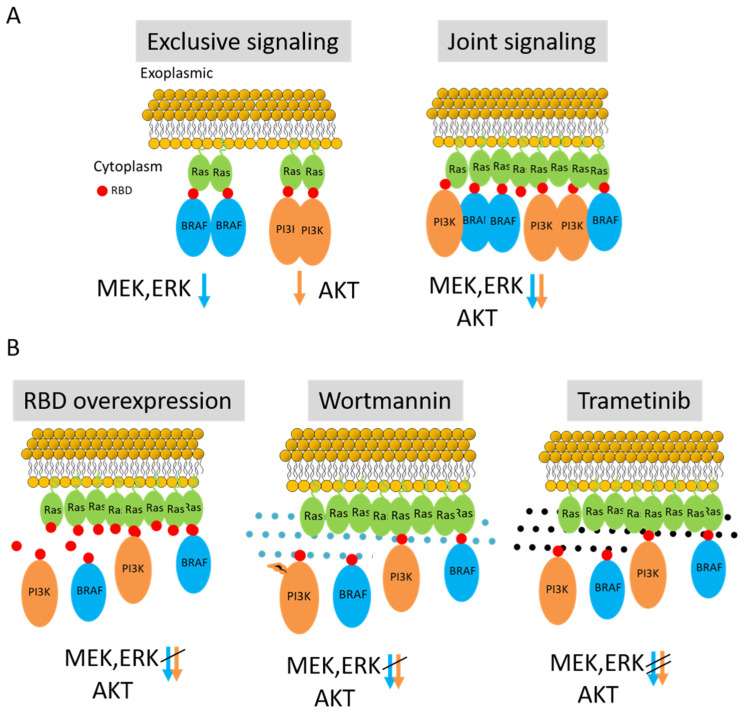
Proposed scheme of signaling in mutual NRas nanoclusters. (**A**) NRas dimers accommodate either BRAF or PI3K, and signal exclusively to either MAPK pathway (MEK, ERK) or to AKT/mTOR (marked by blue and orange arrows, respectively). **Right**, NRas clusters can simultaneously recruit both BRAF and PI3K, signal downstream jointly to MAPK and AKT pathways. (**B**). Recruitment of both PI3K and BRAF to nanoclusters is abrogated by RBD overexpression and competition (**left**), and by treating cells with either wortmannin (**middle**) or trametinib (**right**).

## Data Availability

The original contributions presented in this study are included in the article and Appendix A. Further inquiries can be directed to the corresponding authors.

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
