# Peer review of "NRas Nanoclusters Mediate Crosstalk Between BRAF/ERK and PI3K/AKT Signaling in Melanoma Cells"

_ijms, 2025, doi:10.3390/ijms262311647_

Round 1
Reviewer 1 Report
Comments and Suggestions for Authors
This study provides novel insights into NRAS-mediated signaling in melanoma, highlighting nanoscale colocalization of NRAS, PI3K, and BRAF using advanced single-molecule imaging. The findings are significant, as they reveal unexpected crosstalk between the MAPK and PI3K pathways and suggest potential therapeutic implications. The paper is well-organized, clearly written, and makes a meaningful contribution to the field. I recommend acceptance with minor revisions.
- Many of the microscopy images lack sufficient resolution and clarity. Please provide high-quality versions.
- Significance values (e.g., P-values) should be included in Figure 6C-6Fto demonstrate the statistical differences between treatments.
- The original blot images are too dark to be properly evaluated. Please provide clearer versions.
- The legend for Figure 6 contains the statement, “Graph 0. and < 0.005 respectively of data compared to controls (i.e., either control vector or UT cells).” This phrasing is unclear and should be revised for accuracy and readability.
- In the first paragraph of Discussion, the sentence “Interestingly, we have previously shown that NRas -Q61R also effectively associates with BRAF 15<sup>15</sup><sup>15</sup><sup>15</sup><sup>15</sup><sup>15</sup>.” appears to contain formatting errors and should be corrected.
- Please ensure consistent formatting of citations and reference list.

Reviewer 2 Report
Comments and Suggestions for Authors
In this manuscript, the authors summarized and discussed the nanoscale spatial organization of NRas with its effectors PI3K and BRAF in melanoma cells. The study combines fixed and live-cell two-/three-color super-resolution microscopy (PALM and dSTORM) to demonstrate that NRas forms nanoclusters that simultaneously recruit BRAF and PI3K. These clusters enable crosstalk between the MAPK and PI3K/AKT signaling pathways.
Major Revisions:
- How many cell seeds placed in each well?
- How did control for photobleaching and phototoxicity during live-cell PALM imaging?
- Can the authors clarify whether the segregation of NRas-Q61R from NRas-wt clusters is statistically significant and functionally relevant?
- Could this nanocluster mechanism apply to KRas- or HRas-driven tumors?
Minor Revisions:
1. All sentences must be grammatically correct and clearly written.
For example:
Imaging. We performed two-color photoactivated localization microscopy (PALM) imaging, following protocols established in previous studies (e.g. 14).
Thus, higher values of the PSF [i.e., g(r)] indicate self-clustering.
Still, we propose here another type of crosstalk in which the pathways signals stem from the same protein complexes.(you used different fond)
2. Please ensure that all figure titles and legends are formatted using a uniform font style
Comments on the Quality of English Language
The manuscript would benefit from careful proofreading to correct grammatical errors and improve sentence clarity.
Reviewer 3 Report
Comments and Suggestions for Authors
General assessment
This is a well-executed and timely study that uses advanced single-molecule localization microscopy to explore the nanoscale organization of NRas, PI3K, and BRAF in melanoma cells. The authors convincingly show that NRas nanoclusters act as hubs integrating MAPK and PI3K/AKT signalling, and they demonstrate functional consequences through pharmacological perturbations and proliferation assays.
The novelty of uncovering NRas-driven nanoclusters as dual signalling platforms is high and of broad interest. However, several aspects of methodology, validation, and interpretation could be strengthened to increase the robustness and impact of the study.
Major comments
-
Validation of overexpression artifacts
-
The study relies heavily on fluorescently tagged, overexpressed proteins. While this is a common limitation of PALM, it raises concerns about the physiological relevance of the observed clustering.
-
I encourage the authors to strengthen this point by including or at least discussing possible endogenous validation approaches (e.g., CRISPR knock-in fluorescent tags, or antibody-based SMLM). Even preliminary data or acknowledgment of efforts in this direction would help balance the current reliance on overexpression.
-
-
Functional readouts are limited
-
The data on pAKT, pERK, and proliferation provide a first link between nanoscale organization and cellular behavior. However, the study would benefit from additional functional assays that go beyond cell viability (e.g., apoptosis, cell cycle, migration/invasion).
-
Rescue experiments, for instance with PI3K or BRAF mutants defective in NRas binding, would strengthen the causal interpretation that nanocluster disruption directly drives the observed signaling and proliferation changes.
Minor comments
-
Statistical reporting: Please report the exact n (number of cells) for all quantitative imaging experiments in the main figure legends, not only in the text.
-
Terminology: Sometimes “co-clustering” and “colocalization” are used interchangeably. It would help to define and consistently apply these terms to avoid confusion.
- Figures: Some of the PALM images (e.g., Fig. 1 and Fig. 3) are challenging to interpret due to scale and density. Enlarged insets or schematics could improve readability.
Round 2
Reviewer 2 Report
Comments and Suggestions for Authors
no comments
Author Response
We wish to thank once again the reviewers for their kind consideration and helpful comments on our manuscript.
We have now addressed the remaining minor comment and all required proofreading edits.